# Language Models Can See:
# Plugging Visual Controls in Text Generation

## Abstract

Generative language models (LMs) such as GPT-2/3 can be prompted to generate text with remarkable quality. While they are designed for text-prompted generation, it remains an open question how the generation process could be guided by modalities beyond text such as images. In this work, we propose a training-free framework, called MAGIC (i**MA**ge-**G**uided text generat**I**on with **C**LIP), for plugging in visual controls in the generation process and enabling LMs to perform multimodal tasks (e.g., image captioning) in a zero-shot manner. MAGIC is a simple yet efficient plug-and-play framework, which directly combines an off-the-shelf LM (i.e., GPT-2) and an image-text matching model (i.e., CLIP) for image-grounded text generation. During decoding, MAGIC influences the generation of the LM by introducing a CLIP-induced score, namely *magic score*, which regularizes the generated result to be semantically related to a given image while being coherent to the previously generated context. Notably, the proposed decoding scheme does not involve any gradient update operation, therefore being computationally efficient. On the challenging task of zero-shot image captioning, MAGIC outperforms the state-of-the-art method by notable margins with a nearly 27 times decoding speedup. MAGIC is a flexible framework and is theoretically compatible with any text generation tasks that incorporate image grounding. In the experiments, we showcase that it is also capable of performing visually grounded story generation given both an image and a text prompt.

## 1 Introduction

Since the introduction of GPT-2 (Radford et al., 2019), generative language models (LMs), which are pre-trained on an enormous amount of unstructured text, have produced unmatched performances on a wide range of NLP tasks (Brown et al., 2020; Chowdhery et al., 2022). Given a textual prompt, LMs can continuously generate texts with the next-token prediction decoding scheme. Although controlling the outputs of LMs has become possible by inserting textual prompts, it is still unknown how the decoding process could be guided by information beyond texts, such as images.

Recently, multimodal representation learning of text and images have been rejuvenated by pre-trained image-text joint embedding models, such as CLIP (Radford et al., 2021) and ALIGN (Jia et al., 2021). They leverage large-scale noisy image-text pairs with weak correspondence for contrastive embedding learning and the learned joint model achieves impressive zero-shot performance competitive to supervised models on tasks such as image classification and image-text retrieval. However, they are still under-explored for image-grounded text generation.[1]

How can we combine the best of both the pre-trained LMs and image-text embedding models for visually grounded text generation? Existing supervised methods combine multimodal encoders by further training them on human-annotated paired image-text data (Mokady et al., 2021; Chen et al., 2021a). Differently, weakly supervised approaches (Anderson et al., 2018a; Feng et al., 2019; Laina et al., 2019) rely on pre-trained object detectors to identify visual concepts and create pseudo image-text pairs. Instead of training

---

[1] Note that while such noisy weak image-text pair supervision is sufficient for learning embeddings, they could not be directly used to train image captioning model due to the data's extreme level of noise (Tewel et al., 2021).

on annotated image-text pairs, they directly train on the pseudo data. However, such methods are usually limited by the object detectors that are trained with a fixed set of labels. The closest to our proposal is ZeroCap (Tewel et al., 2021) which is an unsupervised image captioning method by combining frozen CLIP and GPT-2. One of the advantages of ZeroCap is it performs *ex post facto* in the activation space without re-training or fine-tuning the CLIP and GPT-2 models. However, ZeroCap relies on gradient update and optimization over the context cache, which significantly slows down the inference and hinders its use in real-world scenarios.

In this paper, we propose to solve this challenging task in a completely new perspective by designing a novel text decoding scheme, called MAGIC (i**MA**ge-**G**uided text generat**I**on with **C**LIP). During inference, MAGIC does not rely on *any* additional training or parameters and utilizes explicit "control knobs" to select desired outputs following the guidance of both the GPT-2 and CLIP models. Different from the standard decoding process of GPT-2, we insert a CLIP-induced term, namely *magic score*, in the next token search to encourage the predicted result to demonstrate information that is close to the given image. Our experiments show that such a framework enables zero-shot image captioning and also visually grounded story generation under a simple plug-and-play principle.

To verify the qualitative and quantitative performance of the proposed approach, we conduct comprehensive experiments on two commonly used benchmarks (Section §4): MS-COCO (Lin et al., 2014) and Flickr30k (Plummer et al., 2015). To our surprise, MAGIC achieves state-of-the-art (SOTA) performance across different evaluation metrics, especially outperforming all unsupervised and weakly supervised baselines by notable margins. Moreover, since MAGIC involves no gradient update, the inference speed accelerates upon the previous SOTA on zero-shot image captioning by around 27 times. Beyond image captioning, we also test our approach on visually grounded story generation (Section §5). In this task, given an image and a text prompt, MAGIC can generate high-quality stories that outperform strong baseline methods on both human and automatic evaluations.

In summary, we make the following contributions:

- To the best of our knowledge, we are the first to propose a zero-shot method, i.e. MAGIC, to utilize explicit "control knobs" to efficiently select desired outputs following the guidance of both the pre-trained GPT-2 and CLIP models.

- We empirically show that MAGIC is extremely effective on zero-shot image captioning, achieving SOTA across different benchmarks.

- We demonstrate that MAGIC could be used in creative ways: it can perform complex multimodal generation tasks such as visually grounded story generation and reaches near-human performances on a wide range of evaluation metrics.

## 2 Background

In this section, we briefly introduce previous work related to our research.

### 2.1 Image Captioning

Our work is closely related to the literature of image captioning, where the goal is to describe images with meaningful and syntactically correct sentences. Although this topic has been extensively explored in the past few years, it is still far from being considered as a solved task. Given the training strategies (e.g., the type of training data), we can roughly classify the previous methods into two categories: (1) Supervised and (2) Weakly-/Un-Supervised approaches. The former heavily depends on manually labelled image-text datasets. In contrast, the latter tries to create pseudo image-text pairs (i.e., weakly supervised) or even avoid using any paired image-text data (i.e., unsupervised).

**Supervised Approaches.** With the development of deep learning, most of the existing models use one CNN to encode the input image and one RNN to generate the corresponding sentence describing the image (Mao

et al., 2014; Vinyals et al., 2015). These models are trained to maximize the probability of generating the ground-truth captions conditioned on the input image. After that, the main focus of following methods is to model the interaction between visual and textual cues via attention mechanism to get more faithful and richer captions (Xu et al., 2015; Lu et al., 2017; Anderson et al., 2018b; Zhang et al., 2021b; Huang et al., 2021). Meanwhile, some controllable image captioning methods (Mathews et al., 2016; Gan et al., 2017; Chen et al., 2018; Shuster et al., 2019; Chen et al., 2021b) propose to generate diverse descriptions by feeding different control signals (e.g., label and text), which require additional annotations for training. Beyond describing the whole image scene, dense captioning methods (Johnson et al., 2016; Chatterjee & Schwing, 2018; Kim et al., 2019; Yin et al., 2019; Zeng et al., 2020) aim to describe the visual objects in a sub-region of the input image. Recently, vision-language pre-training methods (Zhou et al., 2020; Li et al., 2020; Mokady et al., 2021; Hu et al., 2021), benefiting from the rich visual-textual representation of pre-trained models on large-scale datasets, are tendencies for vision-language generation by re-training or fine-tuning the model parameters on downstream tasks. Although these methods have achieved impressive results, a certain amount of paired image-text data is indispensable during training.

**Weakly-/Un-Supervised Approaches.** Till now, there has been several attempts to reduce the reliance on paired image-text data for the training of image captioning model. In weakly-supervised approaches, employing *pseudo-captions*, i.e., sentences that contain the object labels detected from the given images, has been a common choice (Anderson et al., 2018a; Feng et al., 2019; Laina et al., 2019). However, a weakly supervised cross-modal alignment between image and text is needed. Besides, *pseudo-captions* tend to contain irrelevant words for the given images (Honda et al., 2021). Therefore, it requires carefully designed constraints or learning schema to alleviate the noise. What is more, such methods require a pre-trained object detector bounded by a fixed set of labels to provide visual concepts. They are thus ineffective for any out-of-domain concepts and scenes.

How can we get rid of creating *pseudo-captions* and perform image captioning in a truly unsupervised manner? Recently, CLIP (Radford et al., 2021) has emerged as a successful vision-language alignment model by training on 400M noisy web-collected image-sentence pairs. It has shown impressive zero-shot capabilities on various vision-language tasks and can open new avenues for answering the former question. ZeroCap (Tewel et al., 2021) is the most related to our work. It is built on a pre-trained CLIP model together with the GPT-2 language model (Radford et al., 2019). Different from previous work, ZeroCap is truly zero-shot, where the optimization is performed "*ex post facto*" in the activation space without re-training or fine-tuning the model parameters. In ZeroCap, the whole context cache (i.e., all the $K$ and $V$ in the self-attention modules (Vaswani et al., 2017; Dosovitskiy et al., 2021)) is updated with the guidance of CLIP and GPT-2 for every prediction step. As a result, the computational overhead of such optimization steps will increase drastically as the size of the language model goes up. One key difference of our proposal with respect to ZeroCap is that MAGIC utilizes explicit "control knobs" to select desired outputs corresponding to the given image. Since our procedure does not involve any gradient updating or optimization, it significantly speeds up the decoding process by around 27 times (Section §4.1).

## 2.2 Plug and Play Generative Models

Lagre-scale pre-trained generative models have revolutionized the field of natural language processing (Radford et al., 2019; Brown et al., 2020) and computer vision (Radford et al., 2021; Ramesh et al., 2021; 2022; Karras et al., 2019; 2020; 2021) in the past few years. Various previous work (Nguyen et al., 2016; 2017; Dathathri et al., 2020; Shen et al., 2020) have revealed that there are rich meaningful semantics in the features learned by such models. This shows a promising pathway to synthesize the desired outputs by interpreting the existing generative models in a "*plug and play*" manner. We can then directly enjoy the powerful capabilities of these *off-the-shelf* big models (without any re-training or fine-tuning) and focus on the lightweight task-specific optimization.

For instance, in the image generation field, DGN-AM (Nguyen et al., 2016) can generate images conditioned on a class by finding a hidden code that clearly activates a neuron in another classifier. Then, PPGN (Nguyen et al., 2017) improves the diversity and quality of the synthesized images by incorporating approximate Metropolis-adjusted Langevin (MALA) algorithm (Roberts & Tweedie, 1996; Roberts & Rosenthal, 1998). Shen et al. (2020) propose to directly travel in the latent space of pre-trained unconditional GANs to

manipulate the attributes of the input image. Patashnik et al. (2021) use CLIP to connect the text prompt and images to search the latent codes of StyleGAN by gradient descent optimization, which finally results in the manipulation of the visual attributes in the input image. Similarly, in the text generation field, PPLM (Dathathri et al., 2020) extends the previous PPGN to text generation tasks (i.e., editing topic and sentiment), where the image generative models is replaced with a GPT-2 language model. Most recently, ZeroCap (Tewel et al., 2021) shows image captioning task can be tackled by directly combining the existing CLIP and GPT-2 models. In general, most of these mentioned "plug and play" methods require iteratively shifting the hidden code (or latent code, or context cache) with gradient descent optimization.

Different from previous work, our proposed approach extends the "plug and play" paradigm by optimizing the decoding strategy of generative LMs. Since MAGIC does not involve any gradient update in the inference, it is computationally efficient (e.g., $\sim$27 times faster than ZeroCap). Notably, although GPT-2 could generate synthetic text samples of unprecedented quality, it may not be natural on some task-specific text (Mokady et al., 2021; Shen et al., 2021; Zhang et al., 2021a). Following this observation, we continue fine-tuning the GPT-2 model on the task-specific text corpus in an unsupervised manner to adapt it to the textual domain of the end task (Laina et al., 2019; Honda et al., 2021). The computational consumption of such adaptation is negligible (e.g., less than 2 hours with 1 NVIDIA 1080Ti on MS-COCO). During decoding, the fine-tuned GPT-2 and CLIP models are fixed.

## 3 Methodology

### 3.1 Unsupervised Language Modelling

Following previous studies (Laina et al., 2019; Honda et al., 2021), we first learn an unsupervised language model on the text corpus of the end task to adapt to its textual domain. Typically, given a text sequence $\boldsymbol{x}$, the maximum likelihood estimation (MLE) objective is used to train the language model $\theta$ as

$$\mathcal{L}_{\text{MLE}} = -\frac{1}{|\boldsymbol{x}|} \sum_{i=1}^{|\boldsymbol{x}|} \log p_\theta(x_i|\boldsymbol{x}_{<i}). \tag{1}$$

Recently, Su et al. (2022) propose to incorporate contrastive objective into the training of the language model to calibrate the model's representation space and obtain better language model perplexity. Given the text sequence $\boldsymbol{x}$, the contratsive objective $\mathcal{L}_{\text{CL}}$ is defined as

$$\mathcal{L}_{\text{CL}} = \frac{1}{|\boldsymbol{x}| \times (|\boldsymbol{x}| - 1)} \sum_{i=1}^{|\boldsymbol{x}|} \sum_{j=1, j \neq i}^{|\boldsymbol{x}|} \max\{0, \rho - s(h_{x_i}, h_{x_i}) + s(h_{x_i}, h_{x_j})\}, \tag{2}$$

where $\rho$ is a pre-defined margin that regularizes the distribution of the model's representation space. The $h_{x_i}$ is the representation of token $x_i$ and the similarity function $s$ computes the cosine similarity between token representations as $s(h_{x_i}, h_{x_j}) = h_{x_i}^\top h_{x_j}/(\|h_{x_i}\| \cdot \|h_{x_j}\|)$.

The overall unsupervised learning objective $\mathcal{L}$ of the language model is then defined as

$$\mathcal{L} = \mathcal{L}_{\text{MLE}} + \mathcal{L}_{\text{CL}}. \tag{3}$$

### 3.2 MAGIC Search

We propose a new decoding scheme, *MAGIC Search*, which aims to steer the decoding process of the language model towards a desired visual direction. Formally, given a text prefix $\boldsymbol{x}_{<t}$ and an image $\mathcal{I}$, the selection of the output token $x_t$ at the time step $t$ follows

$$x_t = \underset{v \in V^{(k)}}{\arg\max} \Bigg\{ (1 - \alpha) \times \underbrace{p_\theta(v|\boldsymbol{x}_{<t})}_{\text{model confidence}} -$$

$$\alpha \times \underbrace{(\max\{s(h_v, h_{x_j}) : 1 \leq j \leq t - 1\})}_{\text{degeneration penalty}} + \beta \times \underbrace{f(v|\mathcal{I}, \boldsymbol{x}_{<t}, V^{(k)})}_{\text{magic score}} \Bigg\}, \tag{4}$$

where $V^{(k)}$ is the set of top-$k$ predictions from the model's probability distribution $p_\theta(\cdot|\boldsymbol{x}_{<t})$ and $s$ is described in Section §3.1. $h_v$ is the representation of the candidate token $v$ which is computed by the model given the concatenation of $\boldsymbol{x}_{<t}$ and $v$. Inspired by Su et al. (2022), we incorporate the model confidence and degeneration penalty into Eq. (4) to let the model decode the likely output while avoiding the model degeneration problem.

Meanwhile, we introduce a novel scoring criterion, i.e. *magic score*, to plug in visual controls into the decoding process. Given the candidate $v$, the prefix $\boldsymbol{x}_{<t}$, and the image $\mathcal{I}$, the magic score is defined as the distribution of image-text similarity over the candidate set $V^{(k)}$. We build our image-text similarity measurement with a pre-trained CLIP model and the magic score is then defined as

$$f(v|\mathcal{I}, \boldsymbol{x}_{<t}, V^{(k)}) = \frac{e^{\mathrm{CLIP}(\mathcal{I}, [\boldsymbol{x}_{<t}:v])}}{\sum_{z \in V^{(k)}} e^{\mathrm{CLIP}(\mathcal{I}, [\boldsymbol{x}_{<t}:z])}} = \frac{e^{h_\mathcal{I}^\top h_{[\boldsymbol{x}_{<t}:v]}}}{\sum_{z \in V^{(k)}} e^{h_\mathcal{I}^\top h_{[\boldsymbol{x}_{<t}:z]}}}, \tag{5}$$

where $h_\mathcal{I}$ is the image embedding of $\mathcal{I}$ produced by the CLIP image encoder. The $h_{[\boldsymbol{x}_{<t}:v]}$ is the text embedding of the sequence $[\boldsymbol{x}_{<t}:v]$ produced by the CLIP text encoder and $[:]$ denotes the concatenation operation. Intuitively, the magic score encourages the language model to generate text that is semantically related to the image content and the strength of the visual control is regulated by the hyper-parameter $\beta$ in Eq. (4).

We note that our motivation for using CLIP to align the similarity between partial texts and images is shared with the previous study, i.e. ZeroCap (Tewel et al., 2021).[2] However, in contrast to ZeroCap, MAGIC directly plugs visual controls into the decoding process of the language model, without the need of extra supervised training (Dathathri et al., 2020) or gradient update on additional features (Dathathri et al., 2020; Tewel et al., 2021). This property makes our method much more computationally efficient than previous approaches as demonstrated in our experiments (Section §4.1).

## 4 Zero-Shot Image Captioning

We first evaluate our approach on the task of zero-shot image captioning.

**Evaluation Benchmarks.** We conduct experiments on two widely used benchmarks: MS-COCO (Lin et al., 2014) and Flickr30k (Plummer et al., 2015). For both datasets, we set up the training, validation, and test splits according to the protocols provided by Karpathy & Fei-Fei (2015).

**Implementation Details.** As described in Section §3.1, for each benchmark, we adapt the GPT-2 model on the training text corpus for 3 epochs and the contrastive loss margin $\rho$ in Eq. (3) is set as 0.5. We optimize the model with the Adam optimizer (Kingma & Ba, 2015) and a learning rate of 2e-5. Notably, this fine-tuning procedure is computationally negligible, i.e., less than 2 hours with 1 NVIDIA 1080Ti GPU. During decoding, the generation of the language model starts with a special start-of-sequence (i.e., `[sos]`) token. For MS-COCO, we set the $k$, $\alpha$, and $\beta$ in MAGIC Search (i.e., Eq. (4)) as 45, 0.1, and 2.0 based on the model's performance on the validation set. As for Flickr30k, these values are set as 25, 0.1, and 2.0, respectively.[3]

**Baselines.** We include several zero-shot methods as our baselines. (1) We compare the generated results of the language model by starting from the start-of-sequence (i.e., `[sos]`) token with different decoding methods, including top-$k$ sampling (Fan et al., 2018) with $k = 40$ and nucleus sampling (Holtzman et al., 2020) with $p = 0.95$. Moreover, we include contrastive search (Su et al., 2022) using the same $k$ and $\alpha$ as in MAGIC Search to see the direct effect of the proposed magic score (Eq. (4)).[4] Note that, these methods do **not** take into account the image input, therefore can be used to assess the performance lower-bound of the language model. (2) We also compare with a CLIP-based method, called CLIPRe. Given an image, it retrieves the most related caption from the training text corpus based on the image-text similarity as

---

[2]We acknowledge that CLIP was originally trained to match full texts and images. In Appendix A, we quantitatively show that CLIP also learns to effectively measure the similarity between partial texts and images.

[3]In Appendix B, we provide detailed ablation studies on the effect of different hyper-parameter setups.

[4]Note that, when $\beta$ in Eq. (4) equals to 0, the visual control is disabled and MAGIC Search degenerates to the vanilla contrastive search.

| Model | MS-COCO | | | | | | Flickr30k | | | | | | Speed |
|---|---|---|---|---|---|---|---|---|---|---|---|---|---|
| | B@1 | B@4 | M | R-L | CIDEr | SPICE | B@1 | B@4 | M | R-L | CIDEr | SPICE | |
| *Supervised Approach* | | | | | | | | | | | | | |
| BUTD | 77.2 | 36.2 | 27.0 | 56.4 | 113.5 | 20.3 | - | 27.3 | 21.7 | - | 56.6 | 16.0 | - |
| GVD | - | - | - | - | - | - | 66.9 | 27.3 | 22.5 | - | 62.3 | 16.5 | - |
| UniVLP | - | 36.5 | 28.4 | - | 116.9 | 21.2 | - | 30.1 | 23.0 | - | 67.4 | 17.0 | - |
| ClipCap | - | 33.5 | 27.5 | - | 113.1 | 21.1 | - | - | - | - | - | - | - |
| Oscar | - | 36.5 | 30.3 | - | 123.7 | 23.1 | - | - | - | - | - | - | - |
| LEMON | - | 40.3 | 30.2 | - | 133.3 | 23.3 | - | - | - | - | - | - | - |
| *Weakly Supervised Approach* | | | | | | | | | | | | | |
| UIC | 41.0 | 5.6 | 12.4 | 28.7 | 28.6 | 8.1 | - | - | - | - | - | - | - |
| IC-SME | - | 6.5 | 12.9 | 35.1 | 22.7 | - | - | 7.9 | 13.0 | 32.8 | 9.9 | - | - |
| S2S-SS | 49.5 | 6.3 | 14.0 | 34.5 | 31.9 | 8.6 | - | - | - | - | - | - | - |
| S2S-GCC | 50.4 | 7.6 | 13.5 | 37.3 | 31.8 | 8.4 | - | - | - | - | - | - | - |
| *Unsupervised Approach* | | | | | | | | | | | | | |
| Top-$k$ | 33.6 | 2.4 | 8.3 | 25.6 | 3.8 | 1.7 | 34.0 | 2.9 | 9.0 | 24.4 | 3.3 | 2.7 | 69.9× |
| Nucleus | 32.6 | 2.3 | 7.8 | 24.8 | 3.1 | 1.4 | 32.6 | 2.4 | 8.1 | 23.4 | 2.5 | 2.4 | **72.5×** |
| Contrastive | 39.5 | 3.0 | 10.8 | 30.8 | 7.7 | 2.9 | 37.6 | 4.3 | 9.8 | 25.7 | 8.9 | 4.6 | 50.4× |
| CLIPRe | 39.5 | 4.9 | 11.4 | 29.0 | 13.6 | 5.3 | 38.5 | 5.2 | 11.6 | 27.6 | 10.0 | 5.7 | - |
| ZeroCap | 49.8 | 7.0 | 15.4 | 31.8 | 34.5 | 9.2 | **44.7** | 5.4 | 11.8 | 27.3 | 16.8 | 6.2 | 1.0× |
| MAGIC | **56.8** | **12.9** | **17.4** | **39.9** | **49.3** | **11.3** | 44.5 | **6.4** | **13.1** | **31.6** | **20.4** | **7.1** | 26.6× |

Table 1: Image Captioning Results on MS-COCO and Flickr30k.

measured by CLIP. (3) Lastly, we compare with the current state-of-the-art approach, ZeroCap (Tewel et al., 2021), which performs CLIP-guided gradient update on the language model features during the decoding process. For a fair comparison, we use the **same** language model for ZeroCap as in our approach.

**Evaluation Metrics.** Following the common practice in the literature, we perform evaluation using BLEU-1/4 (B@1/4) (Papineni et al., 2002), METER (M) (Denkowski & Lavie, 2014), ROUGE-L (R-L) (Lin & Och, 2004), CIDEr (Vedantam et al., 2015), and SPICE (Anderson et al., 2016). In addition, we compare the relative decoding speed of our approach against other generation-based baselines. Here, the decoding speed is measured from the average inference time per image instance.[5]

## 4.1 Results

Table 1 shows the results on zero-shot image captioning. For a comprehensive comparison, we also include the results of several representative (1) supervised methods: BUTD (Anderson et al., 2018b), GVD (Zhou et al., 2019), UniVLP (Zhou et al., 2020), ClipCap (Mokady et al., 2021), Oscar (Li et al., 2020), and LEMON (Hu et al., 2021); and (2) weakly supervised methods: UIC (Feng et al., 2019), IC-SME (Laina et al., 2019), S2S-SS and S2S-GCC (Honda et al., 2021).

From the results of Top-$k$, Nucleus, and Contrastive, we see that solely using the unsupervised language model without conditioning on image inputs can hardly generate meaningful captions.[6] On the other hand, the results of CLIPRe show that the ability of measuring image-text similarity enables CLIP to retrieve captions that better correlate with the test image from the training text corpus. However, the performance of CLIPRe still lags behind the current SOTA method, ZeroCap, by a large margin due to the data discrepancy between the training and test sets. Lastly, we observe that, on both benchmarks, MAGIC achieves the best performance on 11 out of 13 metrics, demonstrating the clear advantages of our proposed approach. Notably, while outperforming ZeroCap on 12 out of 13 metrics, MAGIC achieves a nearly 27× decoding speedup. This is because, during the decoding process, MAGIC does not involve any computationally inefficient operations

---

[5]The decoding speed of different methods are measured on the same hardware platform with a batch size of 1.

[6]For stochastic sampling methods (i.e., Top-$k$ and Nucleus), we report the results averaged over 3 runs with different random seeds. We refer to Appendix C for more details on the numerical results.

| Model | MS-COCO ⟹ Flickr30k | | | | | | Flickr30k ⟹ MS-COCO | | | | | |
|---|---|---|---|---|---|---|---|---|---|---|---|---|
| | B@1 | B@4 | M | R-L | CIDEr | SPICE | B@1 | B@4 | M | R-L | CIDEr | SPICE |
| Top-$k$ | 34.9 | 2.4 | 7.5 | 24.2 | 2.3 | 1.7 | 30.0 | 1.8 | 8.5 | 23.6 | 2.5 | 1.7 |
| Nucleus | 33.4 | 1.7 | 7.0 | 23.3 | 1.8 | 1.3 | 29.1 | 1.6 | 8.0 | 22.9 | 2.1 | 1.6 |
| Contrastive | 40.3 | 5.3 | 10.7 | 30.5 | 5.1 | 3.4 | 33.8 | 3.2 | 10.2 | 25.5 | 4.2 | 3.7 |
| CLIPRe | 38.7 | 4.4 | 9.6 | 27.2 | 5.9 | 4.2 | 31.1 | 3.0 | 9.9 | 22.8 | 8.5 | 3.9 |
| MAGIC | **46.4** | **6.2** | **12.2** | **31.3** | **17.5** | **5.9** | **41.4** | **5.2** | **12.5** | **30.7** | **18.3** | **5.7** |

Table 2: Cross-Domain Evaluation. X ⟹ Y means source domain ⟹ target domain.

like gradient updates (Dathathri et al., 2020; Tewel et al., 2021), which further validates the practical usage of our approach.

## 4.2 Cross-Domain Experiment

To test the generalization ability of our approach, we conduct a cross-domain experiment. Specifically, we apply the unsupervised language model adapted to the training text corpus of the source domain (e.g., MS-COCO) to perform inference on the test set of the target domain (e.g., Flickr30k). We compare MAGIC with several zero-shot methods, including Top-$k$, Nucleus, Contrastive, and CLIPRe.[7] For CLIPRe, given a test image from the target domain, it retrieves the most related caption from the training text corpus of the source domain.

Table 2 shows the results on cross-domain evaluation, where we observe performance drops in all methods as compared with the in-domain evaluation results shown in Table 1.[8] Nonetheless, MAGIC still performs the best among all compared methods, demonstrating its clear advantages in terms of robustness and generalization ability.

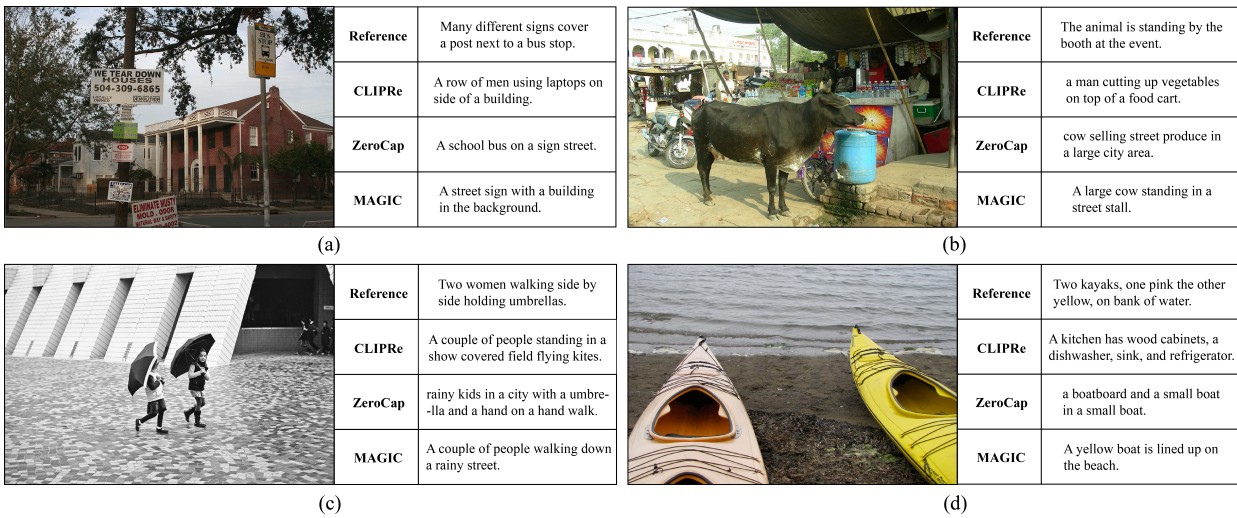

Figure 1: Examples of zero-shot image captioning. (Best viewed by zooming in.)

## 4.3 Qualitative Evaluation

Figure 1 shows visual comparisons between our approach and other two strong zero-shot baselines along with the reference caption.[9] The results demonstrate that MAGIC can generate fluent captions while being more

---

[7]Due to its extremely high computational overhead, we do not include ZeroCap in this experiment.

[8]For stochastic sampling methods (i.e., Top-$k$ and Nucleus), we report the results averaged over 3 runs with different random seeds. We refer to Appendix C for more detailed numerical results.

[9]More examples of zero-shot image captioning are provided in Appendix D.

effective at grounding on the given image. For example, in Figure 1(a), the result of CLIPRe only contains the object "building" that is partially related to the image. As for ZeroCap, it erroneously elaborates the object "school bus" which is not displayed in the image. On the other hand, MAGIC correctly describes the "street sign" object in the image as well as the building in the background. Next, we take Figure 1(d) as an example, in which the result of CLIPRe is clearly irrelevant to the image. As for ZeroCap, while it produces objects like "boatboard" and "small boat" that are related to the image, the generated result is not fluent and ungrammatical. In contrast, MAGIC is able to describe the correct objects such as "yellow boat" and "beach" as well as their positional relationship (i.e., lined up on) while maintaining the fluency and grammaticality of the generated text.

## 5   Story Generation

To verify the versatility and extensibility of MAGIC, we test it on a popular text generation task, i.e., *story generation*. In this task, given a story title (i.e., text prompt), the language model is asked to generate an interesting and coherent story that is related to the story title.

**Evaluation Benchmark.** We evaluate our approach on the widely used ROCStories (Mostafazadeh et al., 2016) dataset. In this dataset, each story title is accompanied with a five-sentence commonsense story written by human. We create the training, validation, and test sets following the official split.

**Model and Baselines.** We use a publicly available GPT-based language model (Su et al., 2022) which is fine-tuned on the ROCStories benchmark.[10] As MAGIC Search is a language model decoding scheme, we compare it with a range of strong text decoding methods, including (1) Greedy search; (2) Beam search with beam width of 10; (3) Top-$k$ sampling (Fan et al., 2018) with $k = 40$; (4) Nucleus sampling (Holtzman et al., 2020) with $p = 0.95$; (5) Typical sampling (Meister et al., 2022) with $\tau = 0.2$; and (6) Contrastive search (Su et al., 2022) with $k = 5$ and $\alpha = 0.6$. The hyperparameters of different methods are selected based on their optimal MAUVE (Pillutla et al., 2021) (detailed in Section §5.1) performance on the validation set.

**Implementations of MAGIC.** To perform MAGIC Search, given the story title, we first retrieve the image (from an image index) that is most related to the story title as measured by CLIP. We construct the image index with the public ConceptualCaptions (Sharma et al., 2018) dataset that contains over 3.3M images from the web. In practice, we pre-compute the image representations with CLIP and build the image index with FAISS (Johnson et al., 2019), therefore supporting a fast "story title-image" retrieval with sub-linear time complexity. Then, by visually grounding on the retrieved image, we generate the story from the story title using MAGIC Search ($k = 5$, $\alpha = 0.6$, and $\beta = 0.15$).[11]

### 5.1   Automatic Evaluation

Following previous studies (Meister et al., 2022; Su et al., 2022), we first evaluate the quality of the generated results from different methods using automatic evaluation metrics, including (1) $n$-gram repetition (rep-$n$); (2) generation diversity (div.); (3) semantic coherence (coh.) between the generated story and the story title; and (4) MAUVE (Pillutla et al., 2021) score that measures the token distribution closeness between the generated text and the human-written text. In addition, to verify that MAGIC is able to generate stories that are semantically related to the given images, we employ CLIPScore (Hessel et al., 2021) to measure the semantic similarity between the generated story and the image retrieved by the story title.

Table 3 shows the automatic results, from which we observe that MAGIC performs the best on most of the evaluation metrics.[12] The results of rep-$n$, diversity, and MAUVE score demonstrate that MAGIC generates the most diverse stories while being closest to human-written stories in terms of token distribution (Pillutla et al., 2021). Moreover, on the coherence (coh.) metric, MAGIC notably outperforms other baselines. We conjecture that the image retrieved by the story title contains rich visual concepts and features, therefore providing more grounding information. As a result, by leveraging these visual knowledge, MAGIC can

---

[10] https://huggingface.co/cambridgeltl/simctg_rocstories

[11] The hyper-parameters are selected based on the model's optimal MAUVE performance on the validation set.

[12] For stochastic methods (i.e., Top-$k$, Nucleus, and Typical sampling), we report the numbers averaged over 3 runs with different random seeds. We refer to Appendix E for more details.

| Method | Automatic Evaluation | | | | | | | Human Evaluation | | | |
|---|---|---|---|---|---|---|---|---|---|---|---|
| | rep-2↓ | rep-3↓ | rep-4↓ | div.↑ | coh.↑ | MAUVE↑ | CLIPScore↑ | coh.↑ | flu.↑ | inform.↑ | si-rel.↑ |
| Agreement | - | - | - | - | - | - | - | 0.68 | 0.57 | 0.66 | 0.73 |
| Greedy | 22.27 | 15.42 | 12.36 | 0.58 | 0.473 | 0.53 | 0.23 | 2.67 | 3.20 | 3.10 | 2.03 |
| Beam | 26.76 | 21.79 | 18.85 | 0.47 | 0.478 | 0.46 | 0.25 | 2.71 | 3.23 | 3.15 | 2.05 |
| Top-$k$ | 3.38 | 0.76 | 0.23 | 0.95 | 0.458 | 0.86 | 0.21 | 2.52 | 3.69 | 3.62 | 1.96 |
| Nucleus | 2.92 | 0.60 | 0.18 | 0.96 | 0.452 | 0.88 | 0.21 | 2.48 | 3.68 | 3.71 | 1.92 |
| Typical | 2.52 | 0.46 | 0.12 | **0.97** | 0.450 | 0.84 | 0.19 | 2.32 | 3.70 | 3.76 | 1.75 |
| Contrastive | **2.49** | **0.38** | **0.09** | 0.97 | 0.488 | 0.89 | 0.28 | 2.86 | 3.72 | 3.76 | 2.07 |
| MAGIC | 2.51 | **0.38** | **0.09** | 0.97 | **0.514** | **0.91** | **0.36** | **3.20** | **3.76** | **3.85** | **2.40** |
| Human | 2.21 | 0.37 | 0.09 | 0.97 | 0.542 | 1.00 | 0.40 | 3.77 | 4.11 | 4.22 | 2.59 |

Table 3: Evaluation results on story generation. ↑ means higher is better and ↓ means lower is better. The best result is **bold** and the second best is underlined. In human evaluation, results significantly outperform the results of other compared methods (Sign Test with p-value < 0.05).

generate stories are more semantically coherent to the story titles. Lastly, on the CLIPScore metric, MAGIC surpasses other methods by large margins, suggesting it generates stories that are more related to the images. In conclusion, the generated text of MAGIC is effectively guided by both the text prompt (i.e., story title) as well as the image, while other methods can only leverage the information from the text prompt.

## 5.2 Human Evaluation

We also conduct a human evaluation with the help of graders proficient in English from a third-party grading platform. We sample 200 instances from the test set. All generated results, plus the reference, are randomly shuffled and evaluated by five graders, resulting in 8,000 annotated samples in total. The evaluation follows a 5-point Likert scale (1, 2, 3, 4, or 5) for each of the following features:[13]

- **Coherence (coh.)**: Whether the generated story is semantically consistent with the title.
- **Fluency (flu.)**: Whether the generated story is fluent and easy to understand.
- **Informativeness (inform.)**: Whether the generated story is diverse and interesting.
- **Story-Image Relevance (si-rel.)**: Whether the generated story is related to the image that is retrieved by the story title.

Table 3 presents the human evaluation results, with the first row showing strong inter-annotator agreements as measured by Fleiss′ kappa coefficient (Fleiss et al., 1971). Firstly, on the fluency (flu.) and informativeness (inform.) metrics, MAGIC performs better than other methods. This indicates that the introduction of visual guidance helps the model to generate more interesting content while maintaining the fluency of the generated story. Moreover, on the coherence metric, the performance gain of MAGIC over other baselines is significant (Sign Test with p-value < 0.05), showing it better maintains the consistency between the generated story and the story title. This conclusion is also validated by the results of coherence score in the automatic evaluation (Section §5.1). Lastly, on the story-image relevance (si-rel.) metric, MAGIC outperforms other methods significantly, demonstrating its ability in generating text by visually grounding on the given image.

## 5.3 Qualitative Evaluation

In Figure 2, we compare our approach with the strongest baseline (i.e., contrastive search), where the image retrieved by the story title is shown on the left-hand side.[14] We see that MAGIC can generate text (highlighted in blue) conditioned on the visual concepts of the image. In the first example, MAGIC elaborates details of three types of ice cream. Such details are more interesting as well as more related to the story title (i.e., ice cream tasting) as compared with the story generated by contrastive search. In the second example, the result of contrastive search is clearly off-the-topic. In contrast, by conditioning on the image,

---

[13]We refer to Appendix F for the detailed human evaluation guidelines.
[14]We refer to Appendix G for more examples of story generation.

| | Title | Ice Cream Tasting |
|---|---|---|
| | **Contrastive** | My family went to a ice cream shop. We ordered the Ice Cream Truck. It was delicious. The customer service was terrible. We had to leave for another day. |
| | **MAGIC** | My family went to a ice cream shop. They ordered three flavors of ice cream. The first one was strawberry, the second was chocolate, and the third was orange. I was excited to try all three flavors. It was very good and I had a great time at the ice cream shop. |
| | Title | Sand Volleyball |
| | **Contrastive** | I went to the park yesterday. It was raining a lot. I had to use the water pump to get to the park. When I got there, there was nothing to play in the park. I ended up playing volleyball instead. |
| | **MAGIC** | I went to the beach with my friends. It was a sand volleyball game. We played for two hours. My friend got to pick his team. He won the game for his team. |

Figure 2: Examples of story generation. MAGIC can generate text (highlighted in blue) that is related to the visual concepts displayed in the image. (Best viewed in color and by zooming in.)

the story generated by MAGIC stays on the topic and describes intriguing details about a sand volleyball game: (i) the game was hosted on the beach; (ii) it lasted for two hours; and (iii) the friend won the game. In summary, by leveraging visual guidance from the image, MAGIC can generate semantically coherent story with interesting content.

## 6 Conclusion and Future Work

In this work, we present MAGIC, a novel decoding scheme that plugs visual controls into the generation of a language model. MAGIC is a training-free framework that enables the LM to address challenging multimodal tasks in a zero-shot manner without sacrificing the decoding speed. To verify the versatility and extensibility of MAGIC, we comprehensively evaluate our approach on two image-grounded text generation tasks: (i) image captioning; and (ii) visually grounded story generation. Experimental results of both automatic and human evaluations demonstrate that our proposed approach outperforms previous state-of-the-art methods by large margins.

**Future work.** While our focus in this study is zero-shot image grounded text generation using a language model, we would like to note that MAGIC Search is a model architecture agnostic decoding scheme. In other words, it can naturally fit into any existing multimodal generative model which takes both the image and text as input. However, it is out of the scope of this paper and we will leave it to future work.

Moreover, in theory, MAGIC is a generic framework that can be extended to modalities beyond text and image. Controls in any form, of any modalities, can be plugged into the language model as long as a certain similarity metric can be found to measure the relevance between the control and the generated text. In future work, we would like to explore the possibility of adapting MAGIC to other modalities beyond images (e.g., audios and videos) therefore enabling the language model to generate text grounded on multimodal intelligence.

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

# Appendix

## Table of Contents

# A  Analysis on the CLIP's Ability in Measuring the Similarity between Partial Texts and Images

In this section, we analyze the ability of CLIP in measuring the similarity between partial texts and images. We first note that the training set of CLIP consists of over 400 million image-text pairs collected from the web. However, this dataset is extremely noisy (Tewel et al., 2021) and it contains a sufficiently large amount of partial and incomplete sentences (Schuhmann et al., 2021). By training on this dataset, CLIP naturally learns to evaluate the similarity between partial texts and images.

For a quantitative evaluation, we conduct an experiment with the following setup. Given an "image-text" pair $(\mathcal{I}, \boldsymbol{x})$, we refer $\boldsymbol{x}$ and $\boldsymbol{x}_{<t}$ to the full captioning text and the partial text (the sequence between index 0 and $t$), respectively, where $t \in \{1, 2, 3, ..., |\boldsymbol{x}|\}$. Then, we define a metric $\text{AD}(t)$ which calculates the absolute difference (AD) between the image-text matching scores computed from the full and partial texts. Specifically, $\text{AD}(t)$ is defined as

$$\text{AD}(t) = \frac{1}{|\mathcal{D}|} \sum_{(\mathcal{I}, \boldsymbol{x}) \in \mathcal{D}} |\text{CLIP}(\mathcal{I}, \boldsymbol{x}) - \text{CLIP}(\mathcal{I}, \boldsymbol{x}_{<t})|, \tag{6}$$

where $\mathcal{D}$ is the set of "image-text" pairs, and $\text{CLIP}(\mathcal{I}, \boldsymbol{x})$ is the CLIP score that indicates the image-text similarity.

For the fairness and robustness of the evaluated results, we randomly sample three sets of "image-text" pairs from the MS-COCO dataset (Lin et al., 2014). In each set, there are 1000 "image-text" pairs, and the average length of the overall sampled texts is 11.85.

| $t$ | 1 | 2 | 3 | 4 | 5 | 6 |
|---|---|---|---|---|---|---|
| $\text{AD}(t)$ | $0.20\,(\pm\textbf{0.08})$ | $0.17\,(\pm\textbf{0.07})$ | $0.13\,(\pm\textbf{0.06})$ | $0.11\,(\pm\textbf{0.04})$ | $0.10\,(\pm\textbf{0.04})$ | $0.08\,(\pm\textbf{0.03})$ |
| $t$ | 7 | 8 | 9 | 10 | 11 | 12 |
| $\text{AD}(t)$ | $0.06\,(\pm\textbf{0.02})$ | $0.05\,(\pm\textbf{0.02})$ | $0.05\,(\pm\textbf{0.02})$ | $0.04\,(\pm\textbf{0.01})$ | $0.01\,(\pm\textbf{0.01})$ | $0.00\,(\pm\textbf{0.00})$ |

Table 4: Results of $\text{AD}(t)$ on MS-COCO.

Table 4 presents the mean and standard deviation of the experimental results. We can see that, as $t$ increases, $\text{AD}(t)$ is inclined to 0. Once the length of the partial text is close to the length of the full text, the CLIP scores are almost the same. Notably, although the length of the partial text has an effect on the CLIP score, its effectiveness is not as large as we expected. For instance, at the beginning (i.e. $t = 1$), the $\text{AD}(t)$ is around 0.2. It indicates that the CLIP score is not very sensitive to the partial text. On the other hand, as $t$ increases, the $\text{AD}(t)$ becomes notably smaller. This suggests that CLIP can effectively align the image and the partial text as long as the **partial** text contains enough content.

Based on the above evaluations, we can conclude that CLIP can effectively measure the "matching score" between partial texts and images which is also the main reason why previous related studies (Tewel et al., 2021; 2022) work well.

# B  Ablation Study on Hyperparameters of MAGIC Search

In this section, we provide further analysis of the effect of hyperparameters in MAGIC Search. To this end, we conduct ablation study experiments on MS-COCO (Lin et al., 2014). Recall from Section §4 that we set $k$, $\alpha$, and $\beta$ in Eq. (4) for MS-COCO as 45, 0.1, 2.0, respectively. To isolate the effect of each hyperparameter, in every experiment, we only vary the value of one hyperparamter while keeping others constant.

**Effect of $k$.** Figure 3(a) shows the performances (i.e., CIDEr and ROUGE-L) of MAGIC Search by varing $k$ from 5 to 55. We observe that, when $k$ is too small (i.e., $k \leq 40$), the performances are not optimal. The reason is that, a small $k$ leads to a too constrained search space, therefore MAGIC Search cannot find the optimal text sequence that best describes the given image. On the other hand, when $k$ is too large (i.e.,

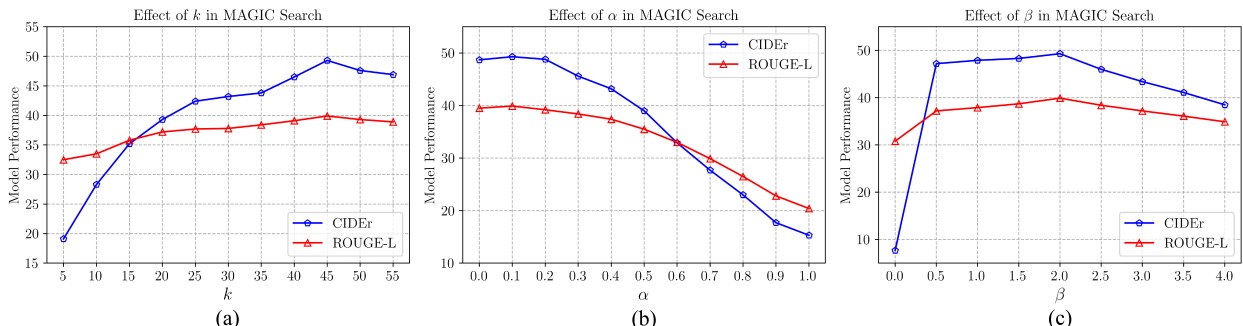

Figure 3: Ablation Study on MS-COCO: (a) Effect of $k$ on MAGIC Search; (b) Effect of $\alpha$ on MAGIC Search; (c) Effect of $\beta$ on MAGIC Search. (Best viewed in color and by zooming in.)

$k \geq 50$), the search space becomes too large therefore introducing extra noise that causes the performance drop in CIDEr and ROUGE-L. In our experiments, the optimal setup for $k$ in MAGIC Search is 45.

**Effect of $\alpha$.** Figure 3(b) demonstrates the performances on MS-COCO by varying $\alpha$ from 0.0 to 1.0. We see that when $\alpha$ is small (i.e., $\alpha \leq 0.2$), the performances are relatively the same. On the other hand, a large $\alpha$ (i.e., $\alpha \geq 0.3$) causes notable drop in the performances. This is due to the fact that a large $\alpha$ forces the language model to generate the text continuation that is less semantically similar to the previously generated context (Su et al., 2022), therefore affecting the performances of MAGIC Search. In our experiments, the optimal setup for $\alpha$ is 0.1.

**Effect of $\beta$.** Lastly, Figure 3(c) illustrates the effect of $\beta$ (from 0.0 to 4.0) on MAGIC Search. Recall from Eq. (4) that, when $\beta = 0.0$, the visual control (i.e., magic score in Eq. (4)) is disabled, therefore MAGIC Search degenerates to the vanilla contrastive search (Su et al., 2022). From the results, we see that, by increasing $\beta$ from 0.0 to 0.5, a significant performance improvement in CIDEr is obtained. Such performance gain clearly demonstrates that the magic score in MAGIC Search is the key factor that enables the language model to generate text grounded on the given image. When $\beta$ is within the range of $[0.5, 2.0]$, the performances of MAGIC Search are relatively the same, indicating the robustness of our approach in terms of the choice of $\beta$. On the other hand, we also see that a large $\beta$ (i.e., $\beta \geq 2.5$) causes the performances to drop, suggesting that the importance of different terms (i.e., model confidence, degeneration penalty, and magic score in Eq. (4)) in MAGIC Search should be properly balanced. In our experiments, the optimal setup for $\beta$ is 2.0.

## C   Detailed Results of Zero-Shot Image Captioning

| Method | run | MS-COCO | | | | | | Flickr30k | | | | | |
|---|---|---|---|---|---|---|---|---|---|---|---|---|---|
| | | B@1 | B@4 | METEOR | R-L | CIDEr | SPICE | B@1 | B@4 | METEOR | R-L | CIDEr | SPICE |
| Top-$k$ | run-1 | 33.8 | 2.4 | 8.4 | 25.7 | 3.9 | 1.8 | 34.1 | 3.1 | 9.0 | 24.4 | 3.3 | 2.8 |
| | run-2 | 33.7 | 2.5 | 8.4 | 25.6 | 4.0 | 1.7 | 34.4 | 2.8 | 9.1 | 24.8 | 3.3 | 2.7 |
| | run-3 | 33.4 | 2.2 | 8.2 | 25.6 | 3.6 | 1.6 | 33.4 | 2.9 | 8.9 | 23.9 | 3.2 | 2.7 |
| | average | 33.6 | 2.4 | 8.3 | 25.6 | 3.8 | 1.7 | 34.0 | 2.9 | 9.0 | 24.4 | 3.3 | 2.7 |
| | std | 0.2 | 0.1 | 0.1 | 0.0 | 0.2 | 0.1 | 0.4 | 0.1 | 0.1 | 0.4 | 0.0 | 0.0 |
| Nucleus | run-1 | 32.6 | 2.3 | 7.8 | 24.8 | 3.2 | 1.5 | 32.5 | 2.5 | 8.4 | 23.5 | 2.7 | 2.4 |
| | run-2 | 32.5 | 2.3 | 7.8 | 24.8 | 3.1 | 1.4 | 32.6 | 2.4 | 8.1 | 23.2 | 2.6 | 2.5 |
| | run-3 | 32.7 | 2.2 | 7.9 | 24.9 | 3.0 | 1.3 | 32.6 | 2.3 | 7.9 | 23.4 | 2.3 | 2.3 |
| | average | 32.6 | 2.3 | 7.8 | 24.8 | 3.1 | 1.4 | 32.6 | 2.4 | 8.1 | 23.4 | 2.5 | 2.4 |
| | std | 0.1 | 0.0 | 0.0 | 0.0 | 0.1 | 0.1 | 0.0 | 0.1 | 0.2 | 0.1 | 0.2 | 0.1 |

**In-domain Result**

| Method | run | MS-COCO $\Longrightarrow$ Flickr30k | | | | | | Flickr30k $\Longrightarrow$ MS-COCO | | | | | |
|---|---|---|---|---|---|---|---|---|---|---|---|---|---|
| | | B@1 | B@4 | METEOR | R-L | CIDEr | SPICE | B@1 | B@4 | METEOR | R-L | CIDEr | SPICE |
| Top-$k$ | run-1 | 34.6 | 2.1 | 7.3 | 24.0 | 2.2 | 1.7 | 29.9 | 1.7 | 8.4 | 23.6 | 2.4 | 1.7 |
| | run-2 | 35.2 | 2.5 | 7.5 | 24.2 | 2.3 | 1.7 | 30.0 | 1.8 | 8.5 | 23.6 | 2.6 | 1.7 |
| | run-3 | 35.0 | 2.6 | 7.6 | 24.5 | 2.5 | 1.8 | 30.0 | 1.8 | 8.5 | 23.6 | 2.6 | 1.7 |
| | average | 34.9 | 2.4 | 7.5 | 24.2 | 2.3 | 1.7 | 30.0 | 1.8 | 8.5 | 23.6 | 2.5 | 1.7 |
| | std | 0.2 | 0.2 | 0.1 | 0.2 | 0.1 | 0.0 | 0.0 | 0.0 | 0.0 | 0.0 | 0.1 | 0.0 |
| Nucleus | run-1 | 33.3 | 1.8 | 7.0 | 23.3 | 1.6 | 1.4 | 29.0 | 1.6 | 8.0 | 22.9 | 2.2 | 1.6 |
| | run-2 | 33.4 | 1.8 | 7.1 | 23.3 | 2.0 | 1.4 | 29.1 | 1.6 | 7.9 | 22.8 | 2.1 | 1.6 |
| | run-3 | 33.5 | 1.5 | 6.9 | 23.4 | 1.9 | 1.2 | 29.1 | 1.6 | 8.0 | 22.9 | 2.0 | 1.5 |
| | average | 33.4 | 1.7 | 7.0 | 23.3 | 1.8 | 1.3 | 29.1 | 1.6 | 8.0 | 22.9 | 2.1 | 1.6 |
| | std | 0.1 | 0.1 | 0.1 | 0.0 | 0.2 | 0.1 | 0.0 | 0.0 | 0.0 | 0.0 | 0.1 | 0.0 |

**Cross-domain Result**

Table 5: Complete numerical results of stochastic methods on zero-shot image captioning. The average and std rows show the mean and standard deviation of results from three different runs.

In Table 5, we show the detailed numerical results of stochastic baselines (i.e., Top-$k$ and Nucleus) on the task of zero-shot image captioning. The upper part of Table 5 presents the results for in-domian experiments (Section §4.1) and the lower part of Table 5 presents the results for cross-domain experiments (Section §4.2). For each method, we report the results of three different runs with different random seeds along with the mean and standard deviation of different runs.

# D    More Visual Examples of Zero-Shot Image Captioning

| | Reference | A vegetarian pizza is half eaten on a pizza holder. |
| | CLIPRe | A chef slides a pizza into a brick oven. |
| | ZeroCap | A pizza with a wooden spoon on a white plate. |
| | MAGIC | Large pizza with vegetables and cheese on a wooden table. |

(a)

| | Reference | A guy squatting and holding a baseball bat at home plate. |
| | CLIPRe | A man with a ball cap and an apron using a brick oven. |
| | ZeroCap | A catcher on the baseball field with a man on the phone. |
| | MAGIC | A baseball player swinging a bat at a ball. |

(b)

| | Reference | A giraffe standing outside of a building next to a tree. |
| | CLIPRe | A man in glasses walks through an open door. |
| | ZeroCap | zoo with large animals in a field of buildings viewable area. |
| | MAGIC | A large giraffe standing in a zoo enclosure. |

(c)

| | Reference | A little girl holding a red frisbee standing on a lush green field. |
| | CLIPRe | A man stands with a knife and onions in front of a garage. |
| | ZeroCap | A baseball player in a swing and a serve. |
| | MAGIC | A child playing with a disc in a backyard. |

(d)

| | Reference | A guy in a jet ski goes fast in a curve. |
| | CLIPRe | A man laying on his stomach with a towel on his head. |
| | ZeroCap | skateboat in motion with a rider on. |
| | MAGIC | A zooming person surfing on a wave in the ocean. |

(e)

| | Reference | A plate with a small square piece of cake with white frosting. |
| | CLIPRe | a close up of a person grabbing a pastry in a container |
| | ZeroCap | chocolate dessert dessert with a fork and a chocolate glaze. |
| | MAGIC | A plate topped with cake and fork. |

(f)

| | Reference | Two birds preparing to eat food off of a plate that was left on a table. |
| | CLIPRe | There are people that are getting food off of the table |
| | ZeroCap | lunch sitting on the tanger island island island |
| | MAGIC | A bird eating bread from a table. |

(g)

| | Reference | A black and white cat is laying on a green pillow on top of a desk. |
| | CLIPRe | A man on his stomach in a white bed. |
| | ZeroCap | stuffed teddy bear sitting in a cat bed with a teddy bear head. |
| | MAGIC | A cat laying on top of a bed. |

(h)

Figure 4: More examples of zero-shot image captioning. (Best viewed by zooming in.)

Figure 4 presents more visual comparisons between our approach against other two strong zero-shot baselines (i.e., CLIPRe and ZeroCap) along with the reference caption.

# E    Detailed Results of Story Generation

Table 6, we present the detailed numerical results of stochastic baselines (i.e., Top-$k$, Nucleus, and Typical) on the task of story generation. For each method, we report the results of three different runs with different random seeds along with the mean and standard deviation of different runs.

# F    Human Evaluation Guidelines

Given the story title and the image, please evaluate the system's result with respect to the following features: (1) Coherence; (2) Fluency; (3) Informativeness; and (4) Story-Image Relevance. In the following, we provide some guidelines regarding how to judge the quality of the system's result in terms of different features.

| Method | run | rep-2↓ | rep-3↓ | rep-4↓ | diversity↑ | coherence↑ | MAUVE↑ | CLIPScore↑ |
|---|---|---|---|---|---|---|---|---|
| Top-$k$ | run-1 | 3.42 | 0.75 | 0.22 | 0.95 | 0.460 | 0.85 | 0.21 |
| | run-2 | 3.27 | 0.77 | 0.25 | 0.95 | 0.455 | 0.87 | 0.21 |
| | run-3 | 3.46 | 0.75 | 0.21 | 0.95 | 0.458 | 0.86 | 0.21 |
| | average | 3.38 | 0.76 | 0.23 | 0.95 | 0.458 | 0.86 | 0.21 |
| | std | 0.08 | 0.01 | 0.02 | 0.00 | 0.002 | 0.01 | 0.00 |
| Nucleus | run-1 | 2.88 | 0.57 | 0.16 | 0.96 | 0.448 | 0.88 | 0.21 |
| | run-2 | 2.88 | 0.61 | 0.18 | 0.96 | 0.454 | 0.87 | 0.22 |
| | run-3 | 2.99 | 0.62 | 0.20 | 0.96 | 0.454 | 0.88 | 0.21 |
| | average | 2.92 | 0.60 | 0.18 | 0.96 | 0.452 | 0.88 | 0.21 |
| | std | 0.05 | 0.02 | 0.02 | 0.00 | 0.003 | 0.00 | 0.00 |
| Typical | run-1 | 2.44 | 0.42 | 0.11 | 0.97 | 0.454 | 0.83 | 0.18 |
| | run-2 | 2.56 | 0.48 | 0.12 | 0.97 | 0.448 | 0.84 | 0.19 |
| | run-3 | 2.55 | 0.48 | 0.14 | 0.97 | 0.448 | 0.85 | 0.19 |
| | average | 2.52 | 0.46 | 0.12 | 0.97 | 0.450 | 0.84 | 0.19 |
| | std | 0.05 | 0.03 | 0.01 | 0.00 | 0.002 | 0.01 | 0.00 |

Table 6: Complete numerical results of stochastic methods on story generation. The average and std rows show the mean and standard deviation of results from three different runs.

### F.1 Coherence

This metric measures whether the system's result is semantically and factually consistent with the story title. The definitions of different scores are:

- [**5**]: The system's result is perfectly in line with the semantic meaning defined by the story title. And all its content is factually supported by or can be logically inferred from the title.

- [**4**]: The system's result is very related to the story title but with some minor errors that does not affect its overall relevance with respect to the story title.

- [**3**]: The system's result is, to some extent, relevant to the story title with some errors that display minor semantic inconsistency or contradiction.

- [**2**]: At the first glance, the system's result seems to be related to the story title. But with careful inspection, the semantic inconsistency can be easily spotted.

- [**1**]: The system's result is obviously off-the-topic or it is semantically contradicted to the content contained in the story title.

### F.2 Fluency

This metric measures the fluency of the system's result. The definitions of different scores are:

- [**5**]: The system's result is human-like, grammatically correct, and very easy to understand.

- [**4**]: Choose this score when you are hesitant between the score 3 and score 5.

- [**3**]: The system's result contains minor errors but they do not affect your understanding.

- [**2**]: Choose this score when you are hesitant between the score 1 and score 3.

- [**1**]: The system's result does not make sense and it is unreadable.

### F.3   Informativeness

This metric measures the diversity, informativeness, and interestingness of the system's result. The definitions of different scores are:

- [**5**]: The system's result is very informative and contains novel content. In addition, it displays a high level of diversity and it is enjoyable to read.

- [**4**]: Choose this score when you are hesitant between the score 3 and score 5.

- [**3**]: The system's result contains some new information and it displays a certain level of diversity.

- [**2**]: Choose this score when you are hesitant between the score 1 and score 3.

- [**1**]: The system's result is dull, repetitive, and does not have new information.

### F.4   Story-Image Relevance

This metric measures the relevance between the system's result with respect to the given image. The definitions of different scores are:

- [**5**]: The system's result is very related to the given image and contains a diverse set of concepts that are visually displayed in the image.

- [**4**]: Choose this score when you are hesitant between the score 3 and score 5.

- [**3**]: The system's result is partially related to the image and some of its content can be found in the image.

- [**2**]: Choose this score when you are hesitant between the score 1 and score 3.

- [**1**]: The system's result is completely irrelevant to the given image.

## G   More Examples of Story Generation

Figure 5 presents more examples generated by contrastive search along with the examples generated by MAGIC Search grounded on the retrieved image. The image retrieved by the story title is shown on the left-hand side of Table 5.

| | Title | Rainstorm |
|---|---|---|
|  | Contrastive | The weatherman predicted a big storm in the future. He went to his house to check on it. There was nothing to see and it was dark. When he woke up, he realized there was no rain. He decided to stay indoors and watch the weather. |
| | MAGIC | The rain started to pour down. I heard a rumble in my yard. It was thundering and heavy. My neighbor came over to see what was happening. He had just bought a big umbrella to protect his house. |
| | Title | French Braid |
|  | Contrastive | The man bought a new scarf. He put it on his head. His wife noticed it was missing. She asked him to look into it. He did not want to look into it. |
| | MAGIC | I wanted to learn a new style of braid. My friend told me I couldn't afford it. I looked online and found some tutorials. After reading all the tutorials, I decided to go for it. It turns out that the best way to learn new braid is to learn French. |
| | Title | The Hair Clump |
|  | Contrastive | The man shaved his head. He went to get a haircut. His hair fell out. The man had to buy new hair. He was happy that he shaved his head. |
| | MAGIC | The hair in my hair was a mess. I went to get some shampoo. After shampooing my hair, it looked better. I decided to keep it that way. Now my hair looks great. |
| | Title | Pig |
|  | Contrastive | The man dug a hole. He saw something in the ground. He asked his neighbors for help. His neighbor helped him dig the hole. The man was happy about his contribution. |
| | MAGIC | The pig came to my door. I put a blanket on it to keep it warm. Then I started playing with it. My neighbor was laughing at me. The pig jumped up and ran away. |

Figure 5: More examples of story generation. MAGIC can generate text (highlighted in blue) that is related to the visual concepts displayed in the image. (Best viewed in color and by zooming in.)

