# OpenReview forum: "Language Models Can See: Plugging Visual Controls in Text Generation"
_TMLR — Withdrawn by Authors_

### Review · Reviewer_ipnk · 2022-12-03

**Summary Of Contributions:**

The work proposes an approach to use language models and separately trained image-text matching models to improve image-grounded text generation. This significantly improves decoding speed and performance and does not require any finetuning of the models on aligned(image+text) data for the task. This work is different from other works which combine LM and image-text models in that it does not require any gradient optimization, which makes it much faster to use for decoding. However, in the current state I have some concerns around some claims in the paper as detailed below. I also suggest a few points to strengthen the paper more.

**Audience:**

Yes

**Claims And Evidence:**

Yes

**Requested Changes:**

I would suggest that the following points are discussed and addressed for acceptance and making it stronger :

1. How would MAGIC score behave for truly off-the-shelf methods i.e. without any fine tuning or adaptation to a specific domain on a specific modality. Otherwise it is not fully accurate to call this a zero-shot method that uses off-the-shelf LMs.  This is particularly highlighted in the abstract as well : "MAGIC is a simple yet efficient plug-and-play framework, which directly combines an off-the-shelf LM (i.e., GPT-2) and an image-text matching model (i.e., CLIP) for image-grounded text generation."
2. More discussion should be added around the relevance of comparing on metrics like CLIPScore/story-image similarity when the images are not ground-truth annotations or there is no proper evaluation set.
3. It will strengthen the paper to compare the Story Generation results with other SOTA methods on that task if available.
4. It will strengthen the paper to add comparison of this approach on other multimodal tasks like Visual Question Answering



**Strengths And Weaknesses:**

## Strengths

1. The method is simple and shows significant improvement over text only methods and ZeroCap. Approach is also significantly faster than ZeroCap during decoding.
2. Authors add detailed ablation on hyper parameters like $\alpha$ and $\beta$ which provides some interesting insights.
3. Thorough results with human evaluation for the experiments on Story Generation on different features like coherence, fluency, informativeness.

## Weaknesses

1. It is first mentioned that the approach can work with off-the-shelf Language models, however from section 3.1, 4 and 5, it is described and observed that the language models needs to be adapted to the end task's target textual domain before applying the decoding method with MAGIC score. Section 4.2 discusses the affect of cross-domain, however the experiments are conducted on image captioning datasets like COCO and Flickr which have some overlap in the visual domains as some of the COCO images are also from Flickr. How does MAGIC perform when the language models are not fine tuned at all (truly off-the-shelf GPT models etc) on the text domain of the end task/text domain of related datasets?
2. For Story Generation, there is no comparison with any other SOTA methods on the task.
3. For Story generation task, the CLIPScore metric as well as the human evaluation about story text-image features are a bit unclear to me. The reason being that the story title and the image retrieved based on CLIP scores are not true ground truth. If we are retrieving the images based on CLIP score and using the same CLIP model for the MAGIC score, it is obvious that the generated text will have more relevance to the retrieved image. How are these types of metric meaningful when comparing with other methods that are not based on any image input. It would have been ideal if there was a human annotated evaluation set that has story title, image and story text to measure the above mentioned metrics on.
4. How does this approach perform if we activate MAGIC score at the beginning of decoding vs later(after waiting k tokens generated)? Does using MAGIC score from the beginning result in bad generations which cannot be recovered from? Is it possible to generate more diverse outputs? Authors discuss about CLIP's ability to measure similarity between partial texts and images in Appendix Section A. However it is not fully clear what is the significance of the AD score in context of the generations.
5. Do the hyperparameters like $\alpha$ and $\beta$ need to be tuned for each task?

---

### Review · Reviewer_RAMk · 2022-12-09

**Summary Of Contributions:**

In this paper, the authors propose to constraint the decoding process of a pretrained language model with pretrained contrastive models like CLIP by modifying the confidence score of decoding certain words with the similarity score between visual input and textual input.

**Audience:**

Yes

**Broader Impact Concerns:**

It will be interesting and somewhat necessary to further evaluate the bias in this model. For example, a further test could be evaluating the model for image captioning and see if it has severe issue of racial bias [r7].

[r7] Understanding and Evaluating Racial Biases in Image Captioning

**Claims And Evidence:**

Yes

**Requested Changes:**

Please see weakness for details.

**Strengths And Weaknesses:**

Strengths:
1. The idea is straightforward and intuitive.
2. Some of the empirical results are strong.

Weakness:
1. Significant lack of discussion of how existing models leverage pretrained language model in a data-efficient manner or visually-constrained decoding [r1-r6]. The literature review should be done more comprehensively and these relevant works should be included for discussion or even comparisons. For example, [r4] adopts another solution to use pretrained LM in a data-efficient manner where the visual inputs are transformed into text descriptions. [r5] and its following-up work has explored quite a lot on visually-constrained decoding. Although they might not explore the same evaluation setting as the proposed method, the representative method like [r5] should have been included for comparisons.
2. The reviewer is still not convinced why the setting is important: zero-shot vision-language tasks with a pretrained language model. Especially considering the strong performance of the in-context learning ability of such pretrained language models, e.g., [r4].


[r1] VX2TEXT: End-to-End Learning of Video-Based Text Generation From Multimodal Inputs, CVPR 2021

[r2] Socratic Models: Composing Zero-Shot Multimodal Reasoning with Language

[r3] Visual Clues: Bridging Vision and Language Foundations for Image Paragraph Captioning

[r4] Language Models with Image Descriptors are Strong Few-Shot Video-Language Learners

[r5] Guided Open Vocabulary Image Captioning with Constrained Beam Search

[r6] Towards Fast Adaptation of Pretrained Contrastive Models for Multi-channel Video-Language Retrieval

---

### Review · Reviewer_RMTN · 2023-01-09

**Summary Of Contributions:**

This paper proposed a training-free framework for image caption generation. More specifically, the authors proposed a new decoding scheme called magic score. Following prior work (Su et al. 2022), which uses contrastive objective as a degeneration penalty, the magic score uses a pre-trained CLIP model to encourage the caption to align with the target image. The proposed method achieve state-of-the-art performance and has 27 times decoding speedup compared to ZeroCap.

**Audience:**

Yes

**Broader Impact Concerns:**

N.A.

**Claims And Evidence:**

Yes

**Requested Changes:**

Besides ZeroCAP, the proposed method is compared with weak baselines that do not incorporate visual information at all. To make the paper more complete, the authors should consider 2 additional baselines to compare.

(1): Socratic type of model [1], that uses CLIP model to predict visual concept and prompt with GPT3 for image captioning.

(2): Constraint decoding type of method such as [2], that uses CLIP model to predict visual concept and decode with GPT2 for zero-shot image captioning,

[1]: Andy Zeng et.al. Socratic Models: Composing Zero-Shot Multimodal Reasoning with Language.

[2]: Lu et.al. NeuroLogic A*esque Decoding: Constrained Text Generation with Lookahead Heuristics.

Note these two additional baselines are very important to benchmark the effectiveness of the proposed CLIP score is better than prompt and constraint decoding approaches.

**Strengths And Weaknesses:**

[Strength]

- Overall, the paper is technically sound and well-written.
- Magic score is a very intuitive method that can combine the generative ability of GPT2 and the image text alignment ability of Clip.
- The proposed method achieves good performance over the prior method and baseline.

[Weakness]

- Magic score is a very intuitive extension of prior work (Su et al. 2022). The technical novelty of the proposed paper is limited, considering the magic score is another form of degeneration penalty given CLIP model.

- Besides ZeroCAP, the proposed method compared with weak baselines that do not incorporate visual information at all. To make the paper more complete, the authors should consider 2 additional baselines to compare.

(1): Socratic type of model [1], that uses CLIP model to predict visual concept and prompt with GPT3 for image captioning.

(2): Constraint decoding type of method such as [2], that uses CLIP model to predict visual concept and decode with GPT2 for zero-shot image captioning,

[1]: Andy Zeng et.al. Socratic Models: Composing Zero-Shot Multimodal Reasoning with Language.

[2]: Lu et.al. NeuroLogic A*esque Decoding: Constrained Text Generation with Lookahead Heuristics.

---

### Note · Authors · 2023-01-09

I have read and agree with the venue's withdrawal policy on behalf of myself and my co-authors.